# On the Redox Equilibrium of TPP/TPPO Containing Cu(I) and Cu(II) Complexes

Stephanie L. Faber, Nesrin I. Dilmen and Sabine Becker *

Fachbereich Chemie, RPTU Kaiserslautern-Landau, Erwin-Schroedinger-Str. 54, 67663 Kaiserslautern, Germany
* Correspondence: sabine.becker@chem.rptu.de; Tel.: +49-(0)631-2055964

**Abstract:** Copper(II) clusters of the type $[Cu^{II}_4OCl_6L_4]$ (L = ligand or solvent) are a well-studied example of inverse coordination compounds. In the past, they have been studied because of their structural, magnetic, and spectroscopic features. They have long been believed to be redox-inactive compounds, but recent findings indicate a complex chemical equilibrium with diverse mononuclear as well as multinuclear copper(I) and copper(II) compounds. Furthermore, depending on the ligand system, such cluster compounds have proven to be versatile catalysts, e.g., in the oxidation of cyclohexane to adipic acid. This report covers a systematic study of the formation of $[Cu^{II}_4OCl_6(TPP)_4]$ and $[Cu^{II}_4OCl_6(TPPO)_4]$ (TPP = triphenylphosphine, $PPh_3$; TPPO = triphenylphosphine oxide, $O=PPh_3$) as well as the redox equilibrium of these compounds with mononuclear copper(I) and copper(II) complexes such as $[Cu^{II}Cl_2(TPPO)_2]$, $[\{Cu^{II}Cl_2\}_2(TPPO)_2]$, $[\{Cu^{II}Cl_2\}_3(TPPO)_2]$, $[\{Cu^{II}_4Cl_4\}(TPP)_4]$, and $[Cu^ICl(TPP)_n]$ (n = 1–3).

**Keywords:** coordination chemistry; copper clusters; inverse coordination; $\mu_4$-oxido copper clusters; TPP; TPPO





## 1. Introduction

Copper(II) clusters of the type $[Cu^{II}_4OCl_6L_4]$ (L = ligand or solvent) are a well-studied example of inverse coordination compounds [1–3]. The first example of these compounds was reported for $[Cu^{II}_4OCl_6(TPPO)_4]$ (TPPO = triphenylphosphine oxide) by Bertrand et al. in 1966 [4]. In the past, such compounds mainly have been investigated regarding their magnetic properties [4–8]. Although these clusters were initially not considered to be of interest concerning their reactivity, recent findings indicate their catalytic activity, for example, in the conversion of cyclohexane to adipic acid [9–13]. However, studies of the reactivity of such compounds can be challenged by the complex chemical equilibrium through which they are linked to other mononuclear and multinuclear compounds [14–16]. With this in mind, we were particularly interested in $[Cu^{II}_4OCl_6(TPP)_4]$ (TPP = triphenylphosphine) and $[Cu^{II}_4OCl_6(TPPO)_4]$. To date, it is unknown if these compounds are also a part of such a complex chemical equilibrium. Yet, it is known that the redox reaction of simple Cu(II) salts with phosphines leads to Cu(I) and TPPO [17–19]. This redox reaction already occurs if traces of $H_2O$ and/or $O_2$ are present and leads to a number of mononuclear and dinuclear copper complexes. E.g., Makáňová et al. described the mononuclear Cu(I) complexes $[Cu^ICl(TPP)_n]$ (n = 1, 2, 3) as well as the Cu(II) complex $[Cu^{II}Cl_2(TPPO)_n)]$ (n = 2, 4) as products of the reaction of $CuCl_2$ with TPP in acetone under inert conditions. Interestingly, $[Cu^{II}_4OCl_6(TPPO)_4]$ was also found in this mixture, which already hints at a complex chemical equilibrium of mononuclear and multinuclear compounds [17]. Both TPP and TPPO are widely used ligands in coordination chemistry. TPP and its derivatives are mainly studied for the luminescent properties of their copper(I) complexes [20–24], which makes the chemical equilibrium with other copper compounds of interest.

In general, the preference of Cu(I) for TPP and Cu(II) for TPPO (as predicted by the HSAB concept) results in a high number of Cu(II)-TPPO and Cu(I)-TPP complexes. In

contrast, few Cu(II)-phosphine complexes are known. In addition to $[Cu^{II}_4OCl_6(TPP)_4]$ [25], one exception is $[Cu^{II}(hfac)_2PR_3]$ with the chelating ligand hfac = hexafluoroacetylacetonate [26].

In this report, a systematic study concerning the formation of $[Cu^{II}_4OCl_6(TPP)_4]$ and $[Cu^{II}_4OCl_6(TPPO)_4]$ is presented. Furthermore, the chemical equilibrium with other Cu(I) and Cu(II) compounds is investigated. Therefore, three synthetic procedures that are known to yield $[Cu^{II}_4OCl_6L_4]$ (L = ligand with amine donor or solvent) have been investigated with regard to the possible formation of $[Cu^{II}_4OCl_6(TPPO)_4]$ and $[Cu^{II}_4OCl_6(TPP)_4]$. Our results establish the complex redox equilibrium of mononuclear and multinuclear Cu(I) and Cu(II) compounds and expands the already known set of TPP- and TPPO-containing compounds that are part of this equilibrium.

## 2. Materials and Methods

Preparation of $[Cu^{II}_4OCl_6(MeOH)_4]$. $[Cu_4^{II}OCl_6(MeOH)_4]$ was prepared according to procedures described in the literature and under inert conditions (under $N_2$ atmosphere, dried solvents) [25]. $CuCl_2$ (6.72 g, 50.0 mmol) and CuO (1.43 mg, 18.0 mmol) were suspended in absolute MeOH (20 mL) and heated to reflux for 5 h. Then, insoluble solid was filtered off the hot reaction mixture. The solvent of the filtrate was removed, which yielded an ochre-colored solid that was dried in vacuum. Yield: 2.47 g (4.03 mmol, 24%).

Preparation of $[Cu^{II}_4OCl_6(CH_3CN)_4]$. Synthesis was carried out under inert conditions. $CuCl_2 \cdot 2H_2O$ (0.60 g, 3.52 mmol, 1.0 eq.), $CuCl_2$ (3.20 g, 23.8 mmol, 6.9 eq.), and sodium tert-butylate (0.75 g, 7.50 mmol, 2.1 eq.) were suspended in absolute $CH_3CN$ (20 mL). After heating to reflux for 2 h, the precipitate was filtered off the hot suspension and discarded. The red filtrate was stored at 4.5 °C. Based on previous experience (yield ca. 90%) the concentration was estimated to be 0.34 mol·$L^{-1}$.

Preparation of $[Cu^{II}_4OCl_6(TPP)_4]$ (**1**). Preparation of **1** was carried out according to procedures described in the literature [25]. Synthesis was carried out under inert conditions. $[Cu^{II}_4OCl_6(MeOH)_4]$ (350 mg, 0.57 mmol, 1 eq.) was dissolved in absolute diethyl ether (50 mL). A solution of TPP (681 mg, 2.60 mmol, 4.6 eq.) in diethyl ether (10 mL) was added dropwise, which yielded a black-brownish precipitate. This solid was filtered off, washed with diethyl ether (10 mL), dried under vacuum, and stored under inert conditions. Yield: 439 mg (0.28 mmol, 50%).

Preparation of 1-crude. $CuCl_2$ (2.22 g, 16.5 mmol, 2.8 eq.), CuO (0.47 g, 5.90 mmol, 1 eq.), and TPP (5.90 g, 22.5 mmol, 4.1 eq.) were suspended in dry methanol (15 mL) under inert conditions and heated to reflux for 3 h. After hot filtration, the filter cake was discarded. Removal of the solvent yielded a dark green solid.

Preparation of **1a–1c**. A fixed amount of $CuCl_2 \cdot 2H_2O$ (100 mg, 0.57 mmol, 1 eq.) was dissolved in acetone (3 mL) under atmospheric conditions and a varying amount of TPP (0.5–4 eq.) in acetone (2–4 mL) was slowly added. After a few minutes, a precipitate formed, which was filtered, washed with a small amount of acetone (6 mL), and dried under vacuum. $[Cu^ICl(TPP)]$ (**1a**): Addition of 0.5 eq. TPP (78 mg, 0.29 mmol). Yield: 13 mg (0.035 mmol, 12%). Alternatively: addition of 1 eq. TPP (153 mg, 0.58 mmol). Yield: 77 mg (0.21 mmol, 37%). $[Cu^ICl(TPP)_2]$ (**1b**): addition of 2 eq. TPP (308 mg, 1.17 mmol). Yield: 235 mg (0.36 mmol, 63%). $[Cu^ICl(TPP)_3]$ (**1c**): addition of 4 eq. TPP (613 mg, 2.35 mmol). Yield: 435 mg of **1c** that contained 1.5 eq. of acetone (0.48 mmol, 84%, corrected yield for pure **1c**: 78%).

Preparation of $[Cu^{II}_4OCl_6(TPPO)_4]$ (**2**). Recrystallization of **2-crude** from acetone via a diethyl ether diffusion yielded red crystals of **2**. The yield was not determined.

Preparation of **2-crude**. $CuCl_2$ (1.11 g, 8.24 mmol, 2.8 eq.), CuO (0.24 g, 2.98 mmol, 1 eq.), and TPPO (3.15 g, 11.3 mmol, 3.8 eq.) were suspended in dry methanol (10 mL) under inert conditions and heated to reflux for 3 h. After filtration of the hot suspension, the solvent was removed from the dark green filtrate, which yielded an ochre-colored solid (2.7 g).

Preparation of [Cu$^{II}$Cl$_2$(TPPO)$_2$] (**2a**). Recrystallization of **1-crude** from acetone via a diethylether diffusion yielded yellow crystals of **2a**. The yield was not determined.

Preparation of [Cu$^{II}$Cl$_2$(TPPO)$_2$]·[{Cu$^{II}$Cl$_2$}$_2$(TPPO)$_2$] (**2a·2b**). We obtained **2a·2b** by recrystallization of **2-crude** via diethyl ether diffusion from CH$_3$CN under atmospheric conditions. Yield: A few green crystals.

Preparation of [{Cu$^{II}$Cl$_2$}$_3$(TPPO)$_2$] (**2c**). Recrystallization of **2-crude** from a mixture of DCM:MeCN (8:1) via diethyl ether diffusion at 4.8 °C yielded a few green crystals of **2c**.

Preparation of [{Cu$^I_4$Cl$_4$}(TPP)$_4$] (**3**). Recrystallization of **1-crude** from acetone via a diethyl ether diffusion under inert conditions yielded colorless crystals of **3**. Yield: 34.0 mg.

## 3. Results

There are three synthetic routes that are known to yield [Cu$^{II}_4$OCl$_6$L$_4$] (L = ligand): (a) template reactions of the *solvento* cluster [Cu$^{II}_4$OCl$_6$(MeOH)$_4$] (MeOH = methanol) with the corresponding ligand L [5,25]; (b) the reaction of CuO, CuCl$_2$, and L in a dry and refluxing solvent, e.g., methanol [25]; and (c) the reaction of CuCl$_2$·2H$_2$O with L in methanol under atmospheric conditions [14,15]. With the exception of the preparation of **1** via a template synthesis, none of these synthetic protocols has been tested for L = TPP/TPPO before. In the following, the results of all three procedures are described for L = TPP/TPPO.

### 3.1. Template Reactions of [Cu$^{II}_4$OCl$_6$(MeOH)$_4$] with TPP/TPPO

The literature describes the preparation of [Cu$^{II}_4$OCl$_6$(TPP)$_4$] (**1**) under inert conditions via the template reaction of the *solvento* cluster [Cu$^{II}_4$OCl$_6$(MeOH)$_4$] with TPP [5,25]. Following this procedure, we were able to obtain **1** from the reaction of the *solvento* cluster [Cu$^{II}_4$OCl$_6$(MeOH)$_4$] with TPP; however, the choice of solvent (and possibly remaining traces of H$_2$O) was of great importance for the success of this synthesis: According to the literature, when [Cu$^{II}_4$OCl$_6$(MeOH)$_4$] was suspended in diethyl ether and TPP (dissolved in diethyl ether) was added, **1** precipitated as amorphous black solid and could be isolated. We additionally repeated this procedure in methanol and, in contrast, when mixing [Cu$^{II}_4$OCl$_6$(MeOH)$_4$] with TPP in methanol, the solution turned dark for a few seconds and then decolorized to form a colorless precipitate, a behavior that has not been described as yet. This precipitate turned out to be a mixture of diverse [Cu$^I$Cl(TPP)n] (n = 1: **1a**, n = 2: **1b**, n = 3: **1c**) complexes. These complexes have been described previously by Makáňová et al.; however, they had obtained them from the reaction of pure CuCl$_2$ with TPP in dry acetone [17]. Similarly, **1c** has been described by Jardine et al., who obtained **1c** by heating CuCl$_2$·2H$_2$O and TPP in ethanol to reflux [27].

We characterized **1** via IR spectroscopy and elemental analysis (see Supplementary Materials). Despite much effort, it was not possible to obtain crystals suitable for X-ray diffraction analysis. Instead, if the diethyl ether solution was stored for a longer time period, colorless crystals formed that were isolated and determined to the already known cubane cluster [{Cu$^I_4$Cl$_4$}(TPP)$_4$] (**3**, Figure 1). Similarly, as for **1a–1c**, the formation of **3** has already been described for the reaction of CuCl$_2$·2H$_2$O with TPP (1:1.5) in hot ethanol [27]. Thus, even though this reactivity has not been observed for clusters of the type [Cu$^{II}_4$OCl$_6$L$_4$], the reduction of Cu(II) to Cu(I) fits with previously described observations in the literature, which describe the reduction of Cu(II) to Cu(I) by an excess of TPP [17–19].

The spectroscopic and spectrometric characteristics of **3** are in accordance with previously reported characteristics and are listed in the Supplementary Materials. Interestingly, few red crystals of **2** also occurred as side product.

To obtain pure **2**, a similar procedure as for the preparation of **1** was applied. Therefore, TPPO was added to a solution of [Cu$^{II}_4$OCl$_6$(CH$_3$CN)$_4$] in CH$_3$CN. However, and despite the solvent used, it was not possible via this attempt to isolate pure **2**. Instead, this attempt led to [Cu$^{II}_4$OCl$_6$(CH$_3$CN)$_{4-n}$(TPPO)$_n$], a mixture of heteroleptic and homoleptic $\mu_4$-oxido clusters with varying amounts of CH$_3$CN/TPPO ligands, which will be reported elsewhere.

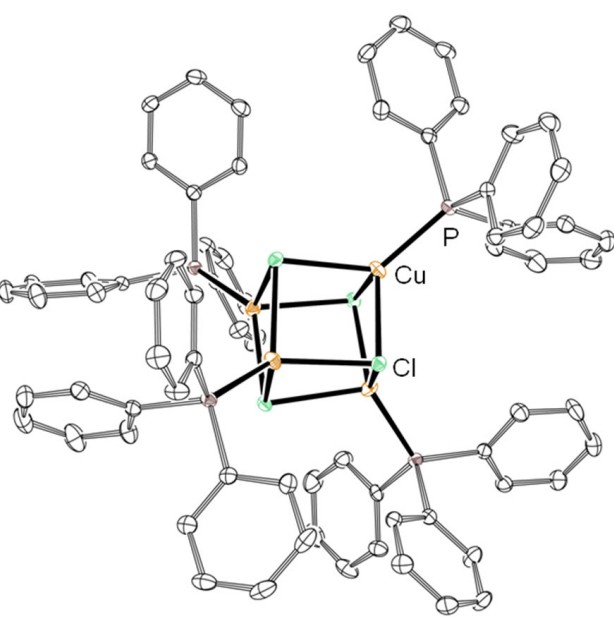

**Figure 1.** Molecular structure of **3**. Thermal ellipsoids at 50%. Hydrogen atoms omitted for clarity.

*3.2. Reaction of CuO, CuCl$_2$, and TPP/TPPO*

Another common synthetic route for the preparation of [Cu$^{II}$$_4$OCl$_6$L$_4$] starts from the simple Cu(II) salts CuO and CuCl$_2$. Here, CuO, CuCl$_2$, and the ligand are suspended in a dry solvent, e.g., methanol, and are heated to reflux [25]. To date, the preparation of either one of **1** or **2** has not been tested via this procedure. To obtain **2**, we mixed CuO, CuCl$_2$, and TPPO in dry methanol and heated it to reflux for 3 h. We obtained an ochre-colored crude product (**2-crude**), from which several Cu(II) complexes could be obtained. For this, we recrystallized **2-crude** from diverse solvents via diethyl ether diffusion. Depending on the solvent used (e.g., acetone, dichloromethane, CH$_3$CN or CHCl$_3$), we obtained different complexes that are described below. When using acetone, red cubic crystals of **2** (Figure 2) were obtained.

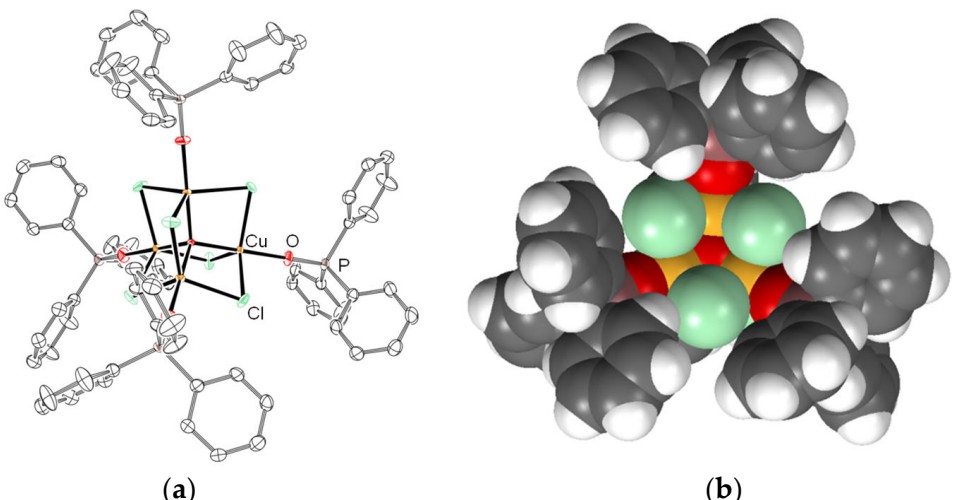

(**a**)                                                                                   (**b**)

**Figure 2.** (**a**) Molecular structure of **2**. Thermal ellipsoids set at 50%; (**b**) space-filling model of **2**.

Crystallization of **2** occurred in the cubic space group $F\bar{4}3c$ and was first described in 1966 by Bertrand and Kelley, who obtained **2** by recrystallization of [Cu$^{II}$Cl$_2$(TPP)$_2$] from methyl isobutyl ketone [4]. The central oxide is inversely coordinated by four Cu(II) centers, yielding a tetrahedron, in which the four Cu(II) centers occupy the corners. Each Cu(II) center is coordinated by three additional Cl$^-$ anions in the equatorial plane and one TPPO

ligand in the axial position, thus generating a trigonal bipyramidal coordination geometry for each Cu(II) center. The parameters determined are in good agreement with parameters already published (see Supplementary Materials) and are not discussed herein. Even though such $\mu_4$-oxido copper clusters are known to be sensitive to moisture [14,25,28,29], **2** can be stored under atmospheric conditions for weeks without being hydrolyzed. In fact, we even observed the formation of **2** under air when mixing $CuCl_2 \cdot 2H_2O$ with TPP (vide infra). One reason for this behavior could be the shielding of the central oxide by the rather bulky TPPO ligands; however, as highlighted by the space-filling model of **2** (Figure 2b), the central oxide remains accessible. Accordingly, the shielding effect by TPPO cannot be the main reason for the comparatively high stability against moisture.

Interestingly, when storing the crystallization solution (acetone) for a couple of weeks, the yellow-colored and already known complex $[Cu^{II}Cl_2(TPPO)_2]$ (**2a**, Figure 3) slowly co-crystallized. Instead of the preferred square-planar coordination geometry for Cu(II) ions, a slightly elongated tetrahedron is observed in **2a**. Depending on the severity of tetragonal distortion, two isomers, α and β, with distinct spectroscopic features, can be distinguished [30,31]. The yellow-colored α-$[Cu^{II}Cl_2(TPPO)_2]$, which also is described within this report, is thermodynamically more stable.

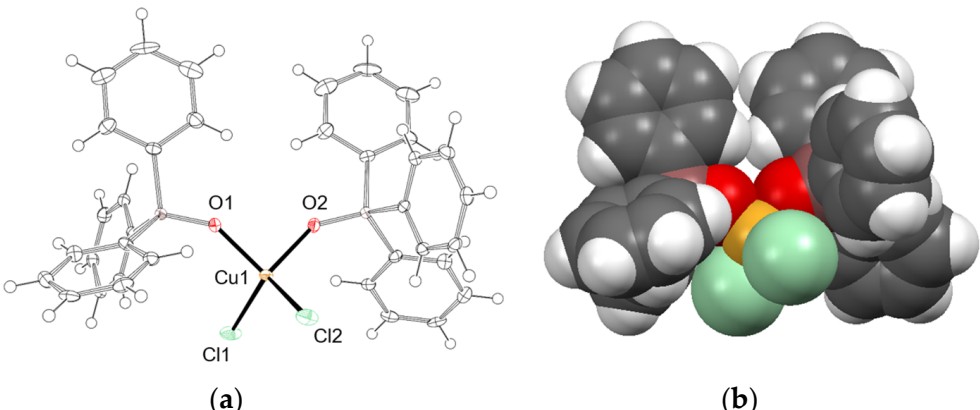

(**a**)  (**b**)

**Figure 3.** (**a**) Molecular structure of **2a**. Thermal ellipsoids set at 50%; (**b**) space-filling model of **2a**. Chosen bond distances: Cu1-O1: 1.9627(19) Å, Cu1-O2: 1.9711(19) Å, Cu1-Cl1: 2.1827(8) Å, Cu1-Cl2: 2.1870(8) Å.

Goodgame and Cotton first obtained **2a** in 1961 and studied its configuration [32]. They obtained **2a** by the reaction of $CuCl_2 \cdot H_2O$ and TPPO in ethanol with subsequent recrystallization from butanone. Additionally, **2a** has been prepared from the reaction of $CuCl_2$ and TPP in an ethanol/acetone mixture at room temperature [17]. The spectroscopic features of **2a** are in accordance with those described in the literature and are listed in the SI.

When using a mixture of $CH_2Cl_2$ or $CHCl_3$ and $CH_3CN$ (ratio 4:1) for the recrystallization of **2-crude**, again, **2a** was obtained. By increasing the amount of $CH_3CN$ in the $CH_2Cl_2/CH_3CN$ solvent mixture, we obtained two similar and as yet unknown compounds, which only differ in the number of {$CuCl_2$} units. Recrystallization of **2-crude** from pure $CH_3CN$ led to green needles of $[Cu^{II}Cl_2(TPPO)_2] \cdot [\{CuCl_2\}_2(TPPO)_2]$ (**2a·2b**). In contrast, a mixture of $CH_2Cl_2/CH_3CN$ (8:1) yielded green needles of $[\{Cu^{II}Cl_2\}_3(TPPO)_2]$ (**2c**, Figure 4).

(**a**)                                                                      (**b**)

**Figure 4.** Molecular structures of **2b** (**a**) and **2c** (**b**). Thermal ellipsoids set at 50%. Chosen bond distances in **2b**: Cu1-O1: 1.9133(19) Å, Cu1-Cl1: 2.1977(8) Å, Cu1-Cl2: 2.2684(8) Å, Cu1-Cl2′: 2.2877(8) Å, Cu1-Cu1′: 3.310(1) Å. Chosen bond distances in **2c**: Cu1-Cl1: 2.2618(6) Å, Cu1-Cl2: 2.3269(6) Å, Cu1-Cl3′: 2.3288(7) Å, Cu1-Cl2′: 2.5490(7) Å, Cu2-Cl2: 2.2535(6) Å, Cu2-Cl3′: 2.2501(6) Å, Cu2-Cl3: 2.2501(6) Å, Cu2-Cl2′: 2.2535(6) Å, Cu1-Cu2: 3.371(1) Å.

Compound **2b** forms individual complex units that pack in a 1:1 ratio with molecules of **2a** in the crystal structure. Both Cu(II) centers show a distorted square-planar coordination geometry, which also is observed for **2a**; however, in **2a**, the distortion is much more pronounced, as indicated by the $\tau_4$ value of 0.59, which even suggests a rather distorted tetrahedral geometry [33]. In **2b**, the $\tau_4$ value is 0.36 and, thus, much closer to the ideal square-planar geometry ($\tau_4 = 0$) than to a tetrahedron ($\tau_4 = 1$). As a consequence of the dinuclearity of **2b**, the Cu-Cl bond distances in the central {Cu$_2$Cl$_2$} diamond are elongated by appr. 3.6% in comparison to the terminal Cu-Cl bond distances. The latter are comparable to the Cu-Cl bond distances in **2a**. Additionally, the Cu-O bonds in **2b** are shorter by appr. 2.8% than in **2a**. In contrast to **2a** and **2b**, **2c** forms zigzag-like chains, in which the [{Cu$^{II}$Cl$_2$}$_3$(TPPO)$_2$] units are linked via the terminal copper centers of the {Cu$_3$Cl$_6$} units (Figure 5). As a result, the terminal Cu(II) centers display a trigonal-bipyramidal coordination geometry with two Cl$^-$ anions and the oxygen atom of TPPO in the equatorial plane. The central Cu(II) center shows a square-planar geometry and lies in the same plane as the other two Cu(II) centers. In general, copper(II) halides are known to form chain-like structures with repeating {CuX$_2$} units; e.g., for CuCl$_2$ and CuBr$_2$, researchers involved in this study have previously reported similar compounds with CH$_3$CN instead of TPPO as ligands [15,16].

To complete our investigations, we repeated the synthetic experiment with TPP instead of TPPO. Thus, instead of mixing CuO and CuCl$_2$ with TPPO, we added TPP and heated the suspension in dry methanol to reflux for 3 h. After filtration and removal of the solvent from the filtrate, a green-colored solid (**1-crude**) was obtained. Recrystallization of **1-crude** from acetone yielded **2a** as the main product. The formation of **2a** can be explained by the redox reaction of the redox pairs Cu(II)/Cu(I) and TPP/TPPO; however, the origin of oxygen is not certain. It is known that traces of H$_2$O in dry solvents leads to the oxidation of TPP to TPPO [17], but at the same time, CuO could have been the oxidant and origin of the oxygen atom. In accordance with the oxidation of TPP to TPPO, the reduction of Cu(II) to Cu(I) was observed: Additionally to **2a**, colorless crystals of CuCl and the new compound CuCl·CH$_3$CN (Figure 6) were found in the sample. In contrast to pure CuCl, CuCl·CH$_3$CN crystallized as a zigzag ladder structure of {Cu$_2$Cl$_2$} diamonds, in which the copper(I) ions are coordinated by three chloride ions. The additional CH$_3$CN ligand completes the tetrahedral coordination geometry. The Cu-Cl bond distances are 2.390(6) Å and thus, slightly elongated in comparison to those of pure CuCl (d(Cu-Cl): 2.339(6) Å). The structure is isostructural with the previously reported structure of CuBr·CH$_3$CN [15].

Interestingly, similar structures containing TPP or triethylphosphine (PEt$_3$) are known; however, [{Cu$^I$Br}$_4$(TPP)$_4$], [{Cu$^I$I}$_4$(TPP)$_4$], and [{Cu$^I$Br}$_4$(PEt$_3$)$_4$] are described as tetramers with a step configuration and not as coordination polymer [18,34–36]. A similar ladder-like structure with phosphine is only known for tetraphosphine ligands that yield octanuclear complexes with CuX (X = Cl, Br, I) [37].

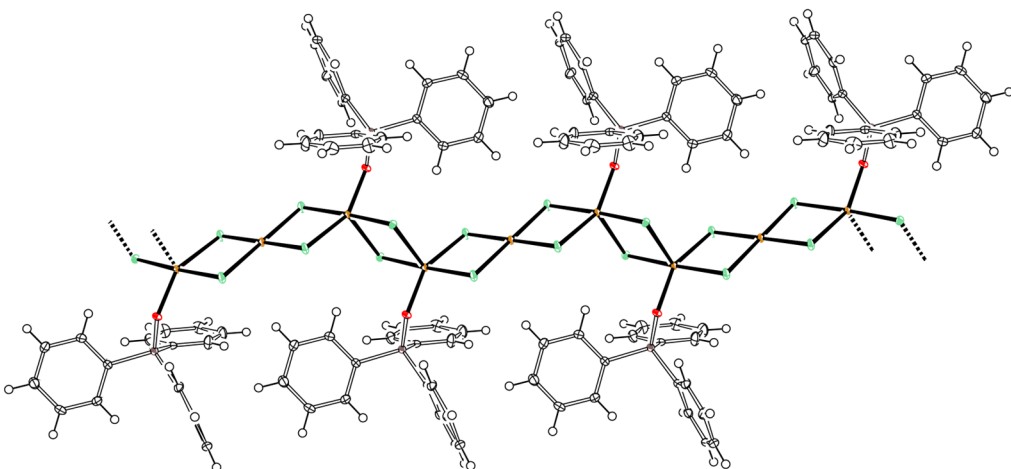

**Figure 5.** Zigzag-like chain that is formed by [{Cu$^{II}$Cl$_2$}$_3$(TPPO)$_2$] units in **2c**. Thermal ellipsoids at 50%.

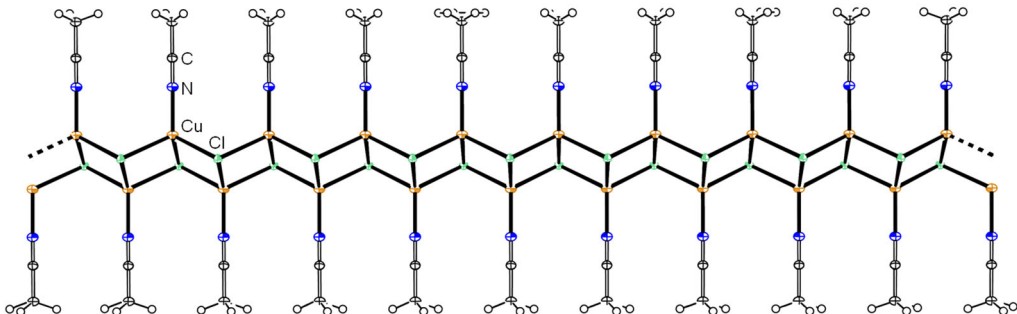

**Figure 6.** Molecular structure of CuCl·CH$_3$CN. Thermal ellipsoids set at 50%.

Interestingly, when refluxing CuO, CuCl$_2$, and TPP in acetone instead of methanol, a gray-colored [Cu$^I$Cl(TPP)$_n$] (n = 1, 2, 3) precipitate also appeared. After separation from the liquid, **3** could be crystallized from the filtrate via diethyl ether diffusion. In general, it is known that even at room temperature, simple Cu(II) and Cu(I) complexes are in a chemical equilibrium with multinuclear species [14,15]. Here, the formation of the main products can be influenced by the stoichiometric ratio of the reactants [14,15]. A similar observation was described by Makáňová et al., who investigated the reactivity of CuCl$_2$ with TPP in acetone with the ratios 1:1 and 1:4 [17]. When using equimolar amounts of CuCl$_2$ and TPP, the Cu(I) compounds [Cu$^I$Cl(TPP)] (**1a**), [Cu$^I$Cl(TPP)$_2$] (**1b**), and [Cu$^I$Cl(TPP)$_3$] (**1c**) were obtained. However, when applying an excess of TPP, the Cu(II) compounds [Cu$^{II}$Cl$_2$(TPPO)$_2$] (**2a**), [Cu$^{II}$Cl$_2$(TPPO)$_4$]·2H$_2$O, and [Cu$^{II}$$_4$OCl$_6$(TPPO)$_4$] (**2**) were obtained. In contrast, in our studies, we could observe a mixture of Cu(II) and Cu(I) compounds without varying the stoichiometric ratios but varying the reaction conditions/starting materials.

### 3.3. Reaction of CuCl$_2$·2H$_2$O and TPP/TPPO under Atmospheric Conditions

It is known that depending on the ligand, [Cu$^{II}$$_4$OCl$_6$L$_4$] can be prepared by simply mixing L with CuCl$_2$·2H$_2$O in methanol [14,15]. Here, it is important to carry out the reaction under atmospheric conditions to provide an oxygen source. Additionally, the stoichiometric ratio of the reactants governs the reaction outcome because of the complex

chemical equilibrium of $[Cu^{II}_4OCl_6(L)_4]$ with simple copper complexes [14,15]. As described above, the outcome of the reaction between $CuCl_2$ and TPP under inert conditions can be influenced by the stoichiometric ratio of the reactants as well [17]. Thus, to complete our studies, we carried out similar reactions to Makáňová et al. In contrast to Makáňová et al., we used non-dry copper(II) chloride, carried out the reactions under atmospheric conditions, and expanded the range of the stoichiometric ratio by the inclusion of the ratios (Cu:TPP) 0.5:1 and 1:2. Therefore, we added TPP in various ratios to a solution of $CuCl_2 \cdot 2H_2O$ in acetone at room temperature. Table 1 summarizes the products that could be isolated from these reaction mixtures.

**Table 1.** Stoichiometric ratios and products isolated from the reaction of $CuCl_2 \cdot 2H_2O$ with TPP under atmospheric conditions in acetone.

| Entry | Cu(II):TPP | Isolated Product | Yield |
|---|---|---|---|
| 1 | 1:0.5 | $[Cu^{II}_4OCl_6(TPPO)_4]$ (**2**) | Few crystals |
| | | $[Cu^ICl(TPP)]$ (**1a**) | 12% |
| 2 | 1:1 | $[Cu^ICl(TPP)]$ (**1a**) | 37% |
| 3 | 1:2 | $[Cu^ICl(TPP)_2]$ (**1b**) | 63% |
| 4 | 1:4 | $[Cu^ICl(TPP)_3]$ (**1c**) | 78% |

Even though our reaction setup differed, the results mainly agree with those described by Makáňová et al. The already described trend that an excess of TPP reduces Cu(II) to Cu(I) could be confirmed, as well [17,18]. When $CuCl_2 \cdot 2H_2O$ was added in excess (Table 1, entry 1), red crystals of **2** occurred as the main product. Additionally, small amounts of $[Cu^ICl(TPP)]$ (**1a**) were obtained as gray precipitate. When using equimolar amounts of $CuCl_2 \cdot 2H_2O$ and TPP (Table 1, entry 2), **1a** was obtained as the main product. In contrast, when using a slight excess of TPP (Table 1, entry 3), a colorless solid precipitated that was identified as $[Cu^ICl(TPP)_2]$ (**1b**) formed. When the amount of TPP was further increased (Table 1, entry 4), $[Cu^ICl(TPP)_3]$ (**1c**) was obtained as colorless precipitate. The remaining filtrate was left for crystallization. Via diethyl ether diffusion, colorless crystals of **1c** could be obtained that were suitable for X-ray diffraction analysis (Figure 7). As known from previous analysis [38,39], the Cu(I) center displays a tetrahedral coordination geometry. Co-crystallization of **1c** occurs with acetone solvent molecules, whose presence corresponds to the traces of acetone in the IR spectrum (see Supplementary Materials) and minor deviations of the elemental analysis (see Supplementary Materials). In general, minor deviations in the elemental analysis results of **1a–1c** are caused by remaining solvent molecules.

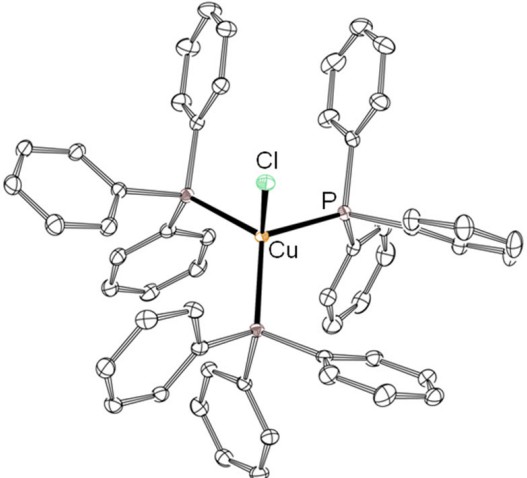

**Figure 7.** Molecular structure of **1c**. Thermal ellipsoids at 50%. Hydrogen atoms and solvent molecules omitted for clarity.

To complete our investigations, we repeated the experiments described above by adding TPPO instead of TPP to the $CuCl_2 \cdot 2H_2O$ solution. Notably, it was only possible to isolate a product when excess of TPPO (1:4) was added. Here, yellow crystals of **2a** and red crystals of **2** appeared. In all other cases, CuCl crystallized as the only product even though the solution colors changed from an initial slightly blue color to brown or orange (ratio 1:4). A change of the solvent to either methanol or ethanol did not lead to any isolable complexes. Even **2** and **2a** could not be isolated from the alcoholic mixtures.

## 4. Discussion

To date, only the reactivity of simple copper salts towards TPP/TPPO had been investigated: it is known that under inert conditions, an excess of TPP reduces $CuCl_2$ to form $[Cu^{I}Cl(TPP)n]$ (n = 1: **1a**, n = 2: **1b**, n = 3: **1c**). Depending on the stoichiometric ratio of the reactants, $[Cu^{II}_4OCl_6(TPPO)_4]$ (**2**) and $[Cu^{II}Cl_2(TPPO)_2]$ (**2a**) were found as by-products [17,18]. However, it is not known to what extent $\mu_4$-oxido copper clusters $[Cu^{II}_4OCl_6L_4]$ behave similarly to simple copper salts and, for example, form multinuclear Cu(I) compounds during the reaction with TPP. Further, it is unknown to what extent multinuclear Cu(I) and Cu(II) species are in equilibrium with the aforementioned, largely mononuclear complexes and, thus, behave analogously to $\mu_4$-oxido copper clusters with amine ligands known from the literature [14–16]. Therefore, building on previous results, we investigated the formation of $[Cu^{II}_4OCl_6(TPP)_4]$ and $[Cu^{II}_4OCl_6(TPPO)_4]$ as well as the reactivity of $[Cu^{II}_4OCl_6L_4]$ (L = MeOH, $CH_3CN$) towards TPP/TPPO. With these experiments, we sought to elucidate the complex redox equilibrium of TPP/TPPO containing Cu(I) and Cu(II) compounds and thus, to complete the set of already known compounds that form this equilibrium. For this purpose, we have chosen three synthetic routes for our investigations, which are known to generally lead to the formation of $\mu_4$-oxido copper clusters with amine donors. To date, only the first approach had been described in the literature as a synthetic route for the preparation of **1**.

In our experiments, we could obtain a huge variety of mononuclear as well as multinuclear copper(II) and copper(I) compounds; these included three new compounds (**2b**, **2c**, CuCl·$CH_3CN$) that could be crystallographically characterized. Furthermore, due to the systematic variation of the reaction conditions, we could show that the reaction between copper(II) chloride and TPP/TPPO yields a complex chemical redox equilibrium of Cu(I)/TPP and Cu(II)/TPPO compounds, which, in contrast to previous studies, does not only include **2** and mononuclear compounds but also multinuclear compounds such as **1**, **2b**, **2c**, and **3**. Furthermore, we could show that the solvent clusters $[Cu^{II}_4OCl_6(solv)_4]$ (solv = methanol, $CH_3CN$) show a similar reactivity towards TPP as that of $CuCl_2$. Depending on the crystallization conditions, individual compounds can be crystallized, as described below.

The template synthesis of $[Cu^{II}_4OCl_6(solv)_4]$ (solv = methanol, $CH_3CN$) yielded **1**, **2**, and **3**. Using acetone instead of methanol and $CH_3CN$ as solvent, pure **3** could be crystallized, thereby confirming our assumption that multinuclear Cu(I) compounds are also involved in the chemical equilibrium. Additionally, $[Cu^{I}Cl(TPP)_n]$ (n = 1, 2, 3) was obtained from the reaction of $[Cu^{II}_4OCl_6(MeOH)_4]$ with TPP in methanol, which, as yet, had only been reported for the reaction of $CuCl_2$ and TPP. It is known that the redox instability of such copper(II) phosphine complexes formed from $CuCl_2$ and TPP results in their decomposition to form Cu(I) complexes [19,25,27], which goes in hand with the reduction of Cu(II) to Cu(I) by an excess of TPP at room temperature [17]. The oxidized product of this reaction is TPPO, which is known to be formed if traces of $H_2O$ are in the reaction solution [17]. The driving force of this reaction is the formation of the P=O bond [40] that is one of the strongest double bonds with 575 kJmol$^{-1}$ bond energy [18,41]. However, in general, the autoxidation of arylphosphines is slow because of the high stability of the intermediate phosphoranyl radical $PH_3\dot{P}O_2R$ [42].

In our experiments, the trend to form Cu(I) compounds was even more pronounced in the reaction of CuO, $CuCl_2$, and TPP under inert conditions at 60 °C, from which **2a**

was obtained as the main product as well as the copper(I) compounds CuCl, CuCl·CH$_3$CN (from CH$_3$CN), and **3** (from acetone); of these Cu(I) compounds, CuCl·CH$_3$CN could be crystallographically characterized for the first time. When carrying out the same reaction with TPPO instead of TPP, a variety of copper(II) compounds (**2**, **2a**, **2b**, and **2c**) were obtained, which extends the known set of CuCl$_2$-TPPO complexes by **2b** and **2c** and highlights the tendency of CuCl$_2$ to form multinuclear complexes that are bridged via chloride ions.

The reaction of CuCl$_2$.2H$_2$O and TPP under atmospheric conditions and at room temperature yielded diverse copper(I) complexes **1a–1c**, which already are known to be formed due to the reaction of CuCl$_2$ and TPP under inert conditions. As to be expected, the number of TPP ligands increased with increasing excess of TPP. Additionally, **2** formed over a long standing time in the reaction mixture with CuCl$_2$.2H$_2$O in excess. Similar studies with TPPO instead of TPP only yielded CuCl and **2a**, whereby the reaction yielding Cu(I) under these conditions is not known. The results of all experiments are presented in Table 2. In general, at least concerning the copper(I) complexes **1a-1c**, indeed, the reaction outcome can be influenced by the stoichiometric ratio of the starting compounds. However, with regard to the copper(II) compounds as well as the multinuclear compounds **1**, **2**, and **3**, the choice of solvent for the preparation as well as for the recrystallization governs the reaction outcome.

**Table 2.** Overview of compounds obtained depending on the reaction conditions.

| Entry | Starting Compound | Ligand Added | Products |
|:---:|:---:|:---:|:---|
| 1 | [Cu$^{II}$$_4$OCl$_6$(MeOH)$_4$] | TPP | [Cu$^{I}$Cl(TPP)$_n$] (n = 1–3, **1a–1c**)<br>[Cu$^{II}$$_4$OCl$_6$(TPP)$_4$] (**1**)<br>[Cu$^{II}$$_4$OCl$_6$(TPPO)$_4$] (**2**)<br>[{Cu$^{I}$$_4$Cl$_4$}(TPP)$_4$] (**3**) |
| 2 | [Cu$^{II}$$_4$OCl$_6$(CH$_3$CN)$_4$] | TPPO | [Cu$^{II}$$_4$OCl$_6$(CH$_3$CN)$_{n-4}$(TPPO)$_n$]<br>[Cu$^{II}$Cl$_2$(TPPO)$_2$] (**2a**) |
| 3 | CuO, CuCl$_2$ | TPP | [{Cu$^{I}$$_4$Cl$_4$}(TPP)$_4$] (**3**)<br>Cu$^{I}$Cl·CH$_3$CN |
| 4 | CuO, CuCl$_2$ | TPPO | [Cu$^{II}$$_4$OCl$_6$(TPPO)$_4$] (**2**)<br>[Cu$^{II}$Cl$_2$(TPPO)$_2$] (**2a**)<br>[{Cu$^{II}$Cl$_2$}$_2$(TPPO)$_2$] (**2b**)<br>[{Cu$^{II}$Cl$_2$}$_3$(TPPO)$_2$] (**2c**) |
| 5 | CuCl$_2$·2H$_2$O | TPP | [Cu$^{II}$$_4$OCl$_6$(TPPO)$_4$] (**2**)<br>[Cu$^{I}$Cl(TPP)] (**1a**)<br>[Cu$^{I}$Cl(TPP)$_2$] (**1b**)<br>[Cu$^{I}$Cl(TPP)$_3$] (**1c**) |
| 6 | CuCl$_2$·2H$_2$O | TPPO | [Cu$^{II}$$_4$OCl$_6$(TPPO)$_4$] (**2**)<br>[Cu$^{II}$Cl$_2$(TPPO)$_2$] (**2a**)<br>CuCl |

We characterized the compounds prepared via IR spectroscopy, melting points, elemental analysis, and, if possible, X-ray diffraction analysis (see Supplementary Materials). Even though only few IR data have been reported to date [25,43,44], IR spectroscopy is a suitable method for the differentiation of the compounds named above. In general, all compounds show similar vibration bands; however, they can be differentiated by the characteristic $\nu$Cu$^{II}$$_4$O, $\nu$Cu$^{I}$$_4$Cl$_4$, $\nu$C-P, and $\nu$P-O in the IR spectra. Compounds **1**, **2**, and **3** display characteristic vibrations of the metal core at 540 cm$^{-1}$, 572 cm$^{-1}$, and 542 cm$^{-1}$, respectively, whereby the values for **1** and **2** correspond well to those described in the literature (see Supplementary Materials Figure S15) [4,17,18,25]. In the free TPPO ligand, $\nu$C-P (542 cm$^{-1}$) is blue-shifted in comparison to the free TPP ligand (513 cm$^{-1}$). Hence, $\nu$C-P of the TPPO-containing compounds **2** (536 cm$^{-1}$) and **2a** (544 cm$^{-1}$) falls into the range of the metal core vibrations of **1** and **3** (see Supplementary Materials, Figure S16).

However, the confusion of these bands, and hence compounds, can be excluded by comparison with the range 1300 cm$^{-1}$–1050 cm$^{-1}$, where only the TPPO-containing compounds **2** and **2a** show the characteristic $\nu$P-O (**2**: 1198 cm$^{-1}$, **2a**: 1143 cm$^{-1}$, free TPPO: 1190 cm$^{-1}$, see Supplementary Materials, Figure S17). Furthermore, **1**, **2**, and **3** show 3 characteristic absorption bands for $\delta$C-H$_{arom.}$ (*out of plane*) in the range 775 cm$^{-1}$–650 cm$^{-1}$, whereas **1a**–**1c** only show 2 characteristic bands (see Supplementary Materials, Figure S18). Again, the TPPO-containing compounds **2** and **2a** as well as free TPPO show three absorption bands (see Supplementary Materials, Figure S19); thus, a clear differentiation of **1**, **1a**–**1c**, **2**, **2a**, and **3** based on IR spectroscopy alone is only possible via the appearance/absence of $\nu$P-O.

Compounds **1a-1c** can be differentiated by the absorptions in the fingerprint region (570–475 cm$^{-1}$) of the IR spectra. The spectra of **1b** and **1c** show vibrations similar to free TPP (517 cm$^{-1}$, 503 cm$^{-1}$, and 494 cm$^{-1}$, TPP: 513 cm$^{-1}$, 500 cm$^{-1}$, and 494 cm$^{-1}$), with the vibration at lowest energy only pronounced as a shoulder. An additional absorption band is observed at 527 cm$^{-1}$. The IR spectrum of **1a**, however, shows only 2 absorption bands, at 542 cm$^{-1}$ and 501 cm$^{-1}$ in this range (see Supplementary Materials Figure S20). Further differences for these compounds are observed in the region 1250 cm$^{-1}$–1050 cm$^{-1}$. Even though all three compounds show similar absorption bands to the free TPP ligand, the shift and intensity of the vibrations are characteristic for **1a**, **1b**, and **1c** (see Supplementary Materials, Figure S21). The IR spectrum of **1c** shows 3 bands of similar intensity at 1221 cm$^{-1}$, 1184 cm$^{-1}$, and 1156 cm$^{-1}$. In contrast, the intensity of the absorption band at 1221 cm$^{-1}$ is much lower in **1a** and **1b**. Furthermore, the spectrum of **1a** shows an additional absorption band at 1121 cm$^{-1}$ that is not observed either for **1b** nor **1c**. Unfortunately, because of the small amounts obtained of **2b** and **2c**, it was not possible to record IR data.

## 5. Conclusions

In this paper, the formation of [Cu$^{II}_4$OCl$_6$(TPP)$_4$] and [Cu$^{II}_4$OCl$_6$(TPPO)$_4$] as well as the reactivity of [Cu$^{II}_4$OCl$_6$(solv)$_4$] (solv = solvent) towards TPP and TPPO was investigated. In addition, the reactivity of CuCl$_2$ ($\cdot$2H$_2$O) towards TPP and TPPO was more extensively reinvestigated. Therefore, the reaction parameters as well as the copper-containing starting compounds were systematically varied. In doing so, previous reports, according to which TPP reduces Cu(II) to Cu(I) to yield [Cu$^I$Cl(TPP)$_n$ (n = 1–3)] (**1a-c**) [17], were also confirmed and could be transferred to the reaction of [Cu$^{II}_4$OCl$_6$(solv)$_4$] with TPP and of CuCl$_2$·2H$_2$O with TPP under atmospheric conditions. In general, our investigations allowed us to show that the previously described mononuclear complexes **1a**–**1c**, [Cu$^{II}$Cl$_2$(TPPO)$_4$]·2H$_2$O, and [Cu$^{II}$Cl$_2$(TPPO)$_2$] are in a chemical redox equilibrium with several multinuclear compounds such as [Cu$^{II}_4$OCl$_6$(TPP)$_4$] (**1**), [Cu$^{II}_4$OCl$_6$(TPPO)$_4$] (**2**), [{Cu$^I_4$Cl$_4$}(TPP)$_4$] (**3**), [{Cu$^{II}$Cl$_2$}$_2$(TPPO)$_2$] (**2b**), and [{Cu$^{II}$Cl$_2$}$_3$(TPPO)$_2$] (**2c**). Additionally, we identified and crystallographically characterized the three compounds **2b**, **2c**, and CuCl·CH$_3$CN for the first time. Taking together our results with a previous study concerning the solution equilibria of tertiary phosphine complexes of CuCl [45], it can be assumed that in solution, all compounds are in equilibrium and are obtained as a complex solid mixture when removing the solvent. The individual compounds can selectively be crystallized from this mixture by varying the solvent for recrystallization. This variation of the solvent and subsequent fractional crystallization could, accordingly, prove to be more suitable for reaction control and product retention than the previous methods, purely based on variation of the stoichiometric ratio of the reactants.

**Supplementary Materials:** The following supporting information can be downloaded at https://www.mdpi.com/article/10.3390/chemistry5020087/s1: Figures S1–S7: Molecular structure of **1c**, **2**, **2a**, **2b**, **2c**, **3** and **CuCl·CH$_3$CN** respectfully; Figures S8–S21: IR spectra, Tables S1–S14: Crystal data and structure refinement and selected bond lengths [Å] and angles [°] for **1c, 2, 2a, 2a·2b, 2c, 3** and **CuCl·CH3CN** respectfully; Table S15: Elemental analysis of **1, 1a, 1b, 1c, 2, 2a** and **3**; Table S16: Melting points of **1a, 1b, 1c, 2,** and **2a**. References [46–52] are cited in the Supplementary Materials.

**Author Contributions:** Conceptualization, S.L.F. and S.B.; validation, S.L.F., N.I.D. and S.B.; investigation, S.L.F. and N.I.D.; writing—original draft preparation, S.L.F.; writing—review and editing, S.B.; visualization, S.L.F. and S.B.; supervision, S.B.; funding acquisition, S.B. All authors have read and agreed to the published version of the manuscript.

**Funding:** This research was funded by the SFB/TRR 88 "3MET–Cooperative effects in homo- and heterometallic complexes".

**Institutional Review Board Statement:** Not applicable.

**Informed Consent Statement:** Not applicable.

**Data Availability Statement:** Additional data (IR spectra, melting points, elemental analyses reports) are presented in the Supplementary Materials. The cif files of all crystal structures have been deposited with the Cambridge Structural Database (CSD). All determined parameters such as bond distances, $U$ij components, etc., can be retrieved free of charge from the CSD (CCDC and CSD numbers: 2255472-2255478).

**Acknowledgments:** The authors thank Jonathan Becker (JLU Gießen) for the crystal measurement.

**Conflicts of Interest:** The authors declare no conflict of interest.

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
