# Peer review of "On the Redox Equilibrium of TPP/TPPO Containing Cu(I) and Cu(II) Complexes"

_chemistry, doi:10.3390/chemistry5020087_

Round 1
Reviewer 1 Report
Becker and co-workers report on a series of CuI and CuII complexes with triphenylphosphine and triphenylphosphine oxide. Honestly speaking, I have been confused because it is not clear to me what is new and what has already been reported in the literature. Thus, I cannot recommend it for publication in Chemistry in its present form. Moreover, I would like to point out the following:
1) Please add the oxidation states of copper in all complexes, for example: [CuII4OCl6(TPP)4].
2) By saying ‘inert conditions’ in the synthetic part what do you mean? Under argon and with dried solvents?
3) Please add a paragraph in the discussion part explaining into great detail what is different (new) in the synthesis of known compounds reported in the manuscript.
4) Please try to make clear to the readers which compounds are new and which structures are new.
Author Response
Becker and co-workers report on a series of CuI and CuII complexes with triphenylphosphine and triphenylphosphine oxide. Honestly speaking, I have been confused because it is not clear to me what is new and what has already been reported in the literature. Thus, I cannot recommend it for publication in Chemistry in its present form. Moreover, I would like to point out the following:
We thank Reviewer 1 for his comment, which helped a lot to improve the quality of the manuscript. We agree that the differentiation of already published results was not clear enough.
To clearly mark which results already had been published already and which ones are new, we re-organized parts of the manuscript, especially the introduction and parts of the results section. Additionally, we added an introductory paragraph into the discussion section that summarizes the results already published and leads on to our investigations. We added phrases such as “so far, it is unknown”, “which has not been described so far”, “the already known compound”, “new compound”, etc. to the description of compounds/procedures to clearly point out the new results.
Furthermore, we added subsections to the manuscript, which organizes the results section into three subsections each dealing with a different synthetic procedure. If necessary, an introductory paragraph was added to these subsections highlighting already existing procedures.
1) Please add the oxidation states of copper in all complexes, for example: [CuII4OCl6(TPP)4].
We thank Reviewer 1 for this comment, which helped to improve the understanding of the redox processes. Accordingly, the oxidation states have been added.
2) By saying ‘inert conditions’ in the synthetic part what do you mean? Under argon and with dried solvents?
Thanks for this question, we carried out the syntheses under an N2 atmosphere and with dried solvents. We added these details in parenthesis to the experimental section.
3) Please add a paragraph in the discussion part explaining into great detail what is different (new) in the synthesis of known compounds reported in the manuscript.
We thank Reviewer 1 for this comment. As stated above, an introductory paragraph highlighting and summarizing the already known results has been added.
4) Please try to make clear to the readers which compounds are new and which structures are new.
Thanks for this comment. As stated above, we re-organized parts of the manuscript to improve the differentiation of new and already published results. In addition, we added phrases such as “already known”, “new compound”, “so far it has been unknown”, etc. to the manuscript.
Reviewer 2 Report
This manuscript by S. Becker and co-workers reports on a systematic study of the formation and reactivity of [Cu4OCl6(PPh3)4] and [Cu4OCl6(O=PPh3)4] copper clusters as well as the complex chemical redox equilibrium of these compounds with mono- and multinuclear Cu(I) and Cu(II) species such as [CuCl2(O=PPh3)2], [{CuCl2}2(O=PPh3)2], [{CuCl2}3(O=PPh3)2], [{Cu4Cl4}(PPh3)4], [CuCl(PPh3)n] (n = 1-3). This paper does significantly complete and value previous works published either by some of the authors and other researchers, see for instance ref. [24]. The various investigated compounds were fully characterized using elemental analysis, FT-IR spectroscopy and single crystal X-ray diffraction as the individual complex has selectively been crystallized from the mixture. The work is competently carried out and the manuscript is well written, presented and organized. Overall, the general quality of this manuscript is high and it deserves to be published in Chemistry as it is.
Minor issues:
Change «at inert/atmospheric/etc conditions» (appearing several times in text) for «under inert/atmospheric/etc conditions»
l. 40: hexaflouroacetylacetonate should be hexafluoroacetylacetonate
Make the reference list consistent by abbreviating all the journal names.
Ref. [12]: provide the international edition of the reference and its DOI.
Ref. [24]: complete the reference with initials of authors #2 and 3, remove «and» between them, put the year after the name of the journal and add the volume and DOI numbers. This reference is not easily accessible to a large readership and it is called many times in text.
Author Response
This manuscript by S. Becker and co-workers reports on a systematic study of the formation and reactivity of [Cu4OCl6(PPh3)4] and [Cu4OCl6(O=PPh3)4] copper clusters as well as the complex chemical redox equilibrium of these compounds with mono- and multinuclear Cu(I) and Cu(II) species such as [CuCl2(O=PPh3)2], [{CuCl2}2(O=PPh3)2], [{CuCl2}3(O=PPh3)2], [{Cu4Cl4}(PPh3)4], [CuCl(PPh3)n] (n = 1-3). This paper does significantly complete and value previous works published either by some of the authors and other researchers, see for instance ref. [24]. The various investigated compounds were fully characterized using elemental analysis, FT-IR spectroscopy and single crystal X-ray diffraction as the individual complex has selectively been crystallized from the mixture. The work is competently carried out and the manuscript is well written, presented and organized. Overall, the general quality of this manuscript is high and it deserves to be published in Chemistry as it is.
We thank Reviewer 2 for his positive feedback and the comments listed below.
Minor issues:
Change «at inert/atmospheric/etc conditions» (appearing several times in text) for «under inert/atmospheric/etc conditions»
Thanks for pointing out the wrong preposition, it has been corrected.
- 40: hexaflouroacetylacetonate should be hexafluoroacetylacetonate
Thanks for pointing out this rather painful typo, it has been corrected.
Make the reference list consistent by abbreviating all the journal names.
Thanks for bringing this to our attention, the reference list has been corrected.
Ref. [12]: provide the international edition of the reference and its DOI.
Thanks for this comment, the international edition is given in reference [11].
Ref. [24]: complete the reference with initials of authors #2 and 3, remove «and» between them, put the year after the name of the journal and add the volume and DOI numbers. This reference is not easily accessible to a large readership and it is called many times in text.
Thanks for pointing out the incomplete information, we corrected it accordingly. Unfortunately, we could not find a DOI number. Caused by the re-organization of the manuscript, reference [24] is now [17].
Round 2
Reviewer 1 Report
The quality of the revised manuscript has been substantially improved and thus, it is my pleasure to recommend it for publication in Chemistry.